# Numerical Analysis of Oxygen-Related Defects in Amorphous In-W-O Nanosheet Thin-Film Transistor

**DOI:** 10.3390/nano11113070

**Published:** 2021-11-15

**Authors:** Wan-Ta Fan, Po-Tsun Liu, Po-Yi Kuo, Chien-Min Chang, I-Han Liu, Yue Kuo

**Affiliations:** 1Department of Photonics and Institute of Electro-Optical Engineering, College of Electrical and Computer Engineering, National Yang Ming Chiao Tung University, Hsinchu 30010, Taiwan; 88capital88@gmail.com (W.-T.F.); eric199303@gmail.com (C.-M.C.); lioujh0324@gmail.com (I.-H.L.); 2Silvaco Taiwan Co., Ltd., Hsinchu 30010, Taiwan; 3Department of Electronic Engineering, Feng Chia University, Taichung 407802, Taiwan; pykuo@fcu.edu.tw; 4Thin Film Nano and Microelectronics Research Laboratory, Texas A&M University, College Station, TX 77843, USA; yuekuo@tamu.edu

**Keywords:** amorphous oxide semiconductor (AOS), density functional theory (DFT), density of states (DOS), high-κ, technology computer aided design (TCAD), thin-film-transistor (TFT)

## Abstract

The integration of 4 nm thick amorphous indium tungsten oxide (a-IWO) and a hafnium oxide (HfO_2_) high-κ gate dielectric has been demonstrated previously as one of promising amorphous oxide semiconductor (AOS) thin-film transistors (TFTs). In this study, the more positive threshold voltage shift (∆V_TH_) and reduced I_ON_ were observed when increasing the oxygen ratio during a-IWO deposition. Through simple material measurements and Technology Computer Aided Design (TCAD) analysis, the distinct correlation between different chemical species and the corresponding bulk and interface density of states (DOS) parameters were systematically deduced, validating the proposed physical mechanisms with a quantum model for a-IWO nanosheet TFT. The effects of oxygen flow on oxygen interstitial (O_i_) defects were numerically proved for modulating bulk dopant concentration N_d_ and interface density of Gaussian acceptor trap N_GA_ at the front channel, significantly dominating the transfer characteristics of a-IWO TFT. Furthermore, based on the studies of density functional theory (DFT) for the correlation between formation energy *E*^f^ of O_i_ defect and Fermi level (*E*_F_) position, we propose a numerical methodology for monitoring the possible concentration distribution of O_i_ as a function of a bias condition for AOS TFTs.

## 1. Introduction

Recently, *n*-type amorphous indium-gallium-zinc-oxide (a-IGZO) [1] has been shown to be one of the most promising materials for amorphous oxide semiconductor (AOS)-based thin-film transistors (TFTs) for achieving a low-temperature process, high-resolution, and a low-power display [2]. The AOS concept indicates that amorphous oxide is composed of heavy metal cations (HMC) with electronic configurations (*n* − 1)d^10^ns^0^ (*n* ≥ 4) [3]. The high-mobility amorphous semiconductors can be achieved because largely spread spherical metal ns^0^ orbitals constitute the lowest unoccupied states (conduction band minimum, CBM), and therefore they are expected to have a high electron mobility and a small electron effective mass in disordered amorphous structures [3]. However, one of the instabilities results from acid-soluble Ga_2_O_3_ and ZnO contained in a-IGZO, which induces back channel damage when etching source/drain electrodes [4]. The role of Ga_2_O_3_ in a-IGZO is to reduce oxygen vacancy (V_O_) for improving the stability of devices [4,5,6,7,8]. Replacing Ga and/or Zn in InO-based semiconductors, such as InTiO, InWO. etc., is an alternative way to suppress V_O_ for reducing instability [4,5,6,7,8], which still maintains the electronic configurations of AOS. In our previous study for enhancing stability [4], we doped a small amount of tungsten oxide (WO_3_) into indium oxide (In_2_O_3_) film to replace gallium oxide (Ga_2_O_3_) from classic IGZO due to the high tungsten-oxide (W-O) bonding-dissociation energy (720 kJ/mol) compared to pure indium oxide (346 kJ/mol) [9].

One of the most important features in amorphous semiconductors is electronic defects, and therefore the defects in either a-IGZO [3,10,11,12,13,14,15] or other AOSs such as amorphous indium tin zinc oxide (a-ITZO) [16] and amorphous tin oxide (a-SnO_x_) [17], etc., have been intensively investigated by theoretical DFT calculations [18,19,20] and experiments. However widely varying process conditions of AOS mainly affect the density of states (DOS) at specific energy levels, corresponding with variations of different chemical species. Additional numerical analysis by Technology Computer Aided Design (TCAD) can be used to understand the physics and material properties in AOS TFTs by studying the effect of the bulk sub-gap density of states (DOS) on electrical characteristics [10,11,12,13,14,15,16,17]. Furthermore, for developing and exploring vast AOS materials, theoretical DFT calculations would be time-exhausting, and utilizing TCAD numerical analysis together with simple material analysis could an alternative and prompt solution to AOS devices.

Furthermore, by using the unique feature of junctionless transistors (JLTs) [21] in CMOS technology, critical control of the dopant profile of source/drain can be avoided to resolve short channel effects (SCEs), such as the instability in threshold voltage (*V_TH_*) as the channel length (L) scales down to a nanometer scale scheme. Additionally, for the gate insulator (GI), certain high-κ materials [22], such as aluminum oxide (Al_2_O_3_) and hafnium oxide (HfO_2_), have been deposited by atomic layer deposition (ALD) [23] with several advantages such as atomic-scaled thickness control, high film density, and superior step coverage and uniformity. Therefore, our previous study demonstrated that the integration of a HfO_2_ GI and ultra-thin amorphous indium tungsten oxide (a-IWO) junctionless TFT can be achieved with promising electrical characteristics, such as high field-effect mobility (*μ_FE_*), near ideal subthreshold swing (*S.S.*) (63 mV/dec), and large ON/OFF current ratios (I_ON_/I_OFF_) (1.5 × 10^8^) [24]. In this work, we investigate experimentally the electrical properties of nanosheet (NS) junctionless a-IWO TFT dependent on different oxygen flows during a-IWO deposition, and then deduce numerically the correlation among the chemical species, materials properties, DOS, and band diagrams by TCAD [25].

## 2. Experiment

The fabricated bottom-metal-gate (BMG) a-IWO TFT is schematically shown in Figure 1a. The molybdenum (Mo) film was deposited as a gate (G) electrode in a direct current (DC) sputter system. The gate channel width (W) and length (L) were defined as 80 μm and 40 μm, respectively, by photolithography and wet etching. Next, a 30 nm thick GI HfO_2_ film was deposited at 250 °C by a plasma enhanced atomic layer deposition (PEALD) system and then annealed in an O_2_ environment for 30 min at 1 atm at 400 °C. Then, a 4 nm a-IWO active semiconducting layer was deposited at room temperature by radio frequency (RF) magnetron sputtering of an In-W-O ceramic plate, containing 96 weight percentage (wt.%) In_2_O_3_ and 4 wt.% WO_3_. The plasma was generated with RF sputtering pressure, power, and radio frequency of 3 mtorr, 50 W, and 13.56 MHz, respectively, to enhance the efficiency of the ion collisions. Table 1 shows that different oxygen ratios were modulated during a-IWO deposition by changing the flux of oxygen (O_2_) and argon (Ar) gas to observe the affected electrical characteristics. The channel patterns were defined by photolithography and dilute hydrofluoric acid (DHF) wet etching. Next, Mo film was deposited in DC sputter and then defined by photolithography and a lift-off process with an overlap length (L_ov_) between the G and S/D of 5 μm. Finally, an organic epoxy-based negative photoresist organic material, SU-8, was deposited by spin coating at a back channel of TFTs as a passivation layer and then annealed at 150 °C for 30 min to ensure a high reliability [8].

## 3. Simulation Methodology

The chemical properties of the a-IWO films dependent on the different oxygen ratios, which were analyzed by X-ray photoelectron spectroscopy (XPS), Instrumentation Center, Taichung City, Taiwan, which can be used to determine qualitatively the bulk DOS of a-IWO for assuring AOS formed with spread spherical metal ns^0^ orbitals in conduction band minimum (CBM) with high electron mobility. To understand the physical mechanisms and materials properties of the a-IWO TFT, TCAD was used for numerical analysis. Measured physical quantities such as layer thicknesses, bandgap (*Eg*), and dielectric permittivity (*ε*) of a-IWO and HfO_2_ were input as the simulation parameters to accurately analyze the physical mechanisms. The bandgaps (*Eg*) of a-IWO and HfO_2_ were 3.05 eV and 5.70 eV, respectively. The electron affinity of a-IWO was estimated as 4.30 eV by the linear relation (1). Schottky barrier work function (*Φ_SD_*) at S/D and Gate electrode’s work function (*Φ_G_*) were 4.67 eV and 4.80 eV, respectively. The carrier concentration *n* and *n*-type doping concentration *N*_d_ of a-IWO determined by Hall effect measurement as inputs for more accurate simulation. We initially assumed *N_d_* was 8 × 10^18^ cm^−3^ for a 3% oxygen ratio of a-IWO film; then, the other corresponding *N*_d_ could be deduced for other oxygen ratios of a-IWO.
χ_IWO_ = 96%·χ_In2O3_ + 4%·χ_WO3_(1)

Based on the intensively investigated electronic defects in AOSs [3,18,19], especially for IGZO structures, other AOS TFTs, such as a-SnOx, a-ITZO, etc., have also been numerically analyzed by characterizing the DOS [16,17]. Therefore, in this study, the adopted physical models and materials properties of a-IWO TFT were fundamentally inherited from a-IGZO publications and then corrected to some extent by materials measurements and TCAD calibration or assumptions [10,11,12,13,14,15,16,17] from measured transfer characteristics. The physical models contain Maxwell–Boltzmann statistics, the drift-diffusion (DD) model, Poisson’s equation, and carrier continuity equations with Shockley–Read–Hall (SRH) recombination. The Bohm quantum potential (BQP) model [26] was enabled for obtaining accurate space charge density in an ultra-thin semiconductor. A Schottky tunneling model was used at the S/D metal–semiconductor (MS) interface [27]. The electron mobility model was used from a-IGZO dependent on electron concentration *n* and lattice temperature *T_L_* in Equations (2) and (3) [28].
*μ_n_* = *μ*_*n*0_ (*n*/*n_crit_*) ^*γ*/2^
(2)
*γ* = *γ*_0_ + *T**_γ_*/*T_L_*(3)
where *μ_n_*_0_ is the intrinsic electron mobility; in this study, *μ_n_*_0_ was based on the experimentally extracted field-effect mobility (*μ_FE_*). *n* is the electron concentration, which can refer to the Hall measurement. Default critical electron concentration *n_crit_* was 1 × 10^20^ cm^−3^. In this study, the temperature coefficient of electron mobility *γ* was determined by setting the intrinsic temperature coefficient *γ*_0_, lattice temperature *T_L_* and temperature coefficient *T_γ_* as −0.36, 300 K and 178.4 K, respectively [28].

As for one of the basic semiconductor equations is Poisson’s equation relating the electrostatic potential (*φ*) to the space charge density (*ρ*) and is given by
∇·(*ε* ∇*φ*) = −*ρ* = −*q*(*p_free_* − *n_free_* + *n_T_* − *p_T_* + *N_d_*)(4)
where *ε* is the local permittivity of a-IWO, and HfO_2_ was set to 10 and 18, respectively. *q* is the elementary charge, *p_free_* and *n_free_* are the free hole and electron density, respectively, and the ionized densities of acceptor-like and donor-like trap (*n_T_* and *p_T_,* respectively) are given in Equations (5) and (6), which are integrated from valance band edge *E_V_* to conduction band edge *E_C_*.

The densities of ionized traps relate the occupation probability *f*(*E*) of a trap level at energy *E* in Equations (7) and (8) for the acceptor and donor traps, respectively, which are dependent on the intrinsic carrier concentration *n_i_*, intrinsic Fermi level *E*_i_, lattice temperature *T*_L_, capture cross sections (*σ_E_* and *σ_H_*), and the thermal velocity (*υ_n_* and *υ_p_*) for the electron and hole. The occupation probability *f*(*E*) describes that traps are either filled with electrons or empty; then, it has a value in the range of 0 to 1, and consequently the Fermi energy position can be determined when *f*(*E*) = 0.5. The occupation probability *f*(*E*) and thermal velocity *υ_n_* and *υ_p_* for the electron and hole in Equations (9) and (10) can be determined by assuming the Richardson coefficients (*A_n_*, *A_p_*) of 41 A·cm^−2^·k^−2^ [29], capture cross sections *σ_E_* and *σ_H_* of 1 × 10^12^ cm^2^ [30], and conduction/valance band effective density of states (*N_C_* = *N_V_*) of 2.0 × 10^18^ cm^−3^, initially estimated from the carrier concentration of Hall measurements.
*E*_C_*n*_T_ = ∫ *g_A_*(*E*) *f_A_*(*E*, *n*, *p*) *dE*(5)
*E*_V_*E*_C_*p*_T_ = ∫ *g_D_*(*E*) *f_D_*(*E*, *n*, *p*) *dE*(6)
*E*_V_*f_A_*(*E*, *n*, *p*) = {*ν_p_σ_H_·n_i_·exp*[(*E_i_* − *E*)/*kT_L_*] + *ν_n_σ_E_·n*}/{*ν_p_σ_H_*[*p* + *n_i_·exp*[(*E_i_* − *E*)/*kT_L_*] + *ν_n_σ_E_*[*n* + *n_i_·exp*[(*E* − *E_i_*)/*kT_L_*}(7)
*f_D_*(*E*, *n*, *p*) = 1 − *f_A_*(*E*, *n*, *p*)(8)
*ν_n_* = *A_n_*·*τ*^2^/(*qN_c_*)(9)
*ν_p_* = *A_n_*·*τ*^2^/(*qN_c_*)(10)
*g*(*E*) = *g_A_*(*E*) + *g_D_*(*E*) = *g_TA_*(*E*) + *g_GA_*(*E*) + *g_TD_*(*E*) + *g_GD_*(*E*)(11)

Equation (11) shows that the density of state (DOS) *g*(*E*) of AOS is composed of an acceptor-like trap *g_A_*(*E*) and a donor-like trap *g_D_*(*E*), with a combination of exponential tail and Gaussian distributions. In this study, we assumed the chemical properties between a-IWO and a-IGZO were similar, and then the schematic DOS of a-IGZO represented in Figure 1b could be used as a starting point to simulate the a-IWO TFT. In order to understand that each DOS distribution stands for the corresponding chemical species in AOS film, in this study we defined different mid-gap DOS distributions as control variables for examining the effects on electrical characteristics in later sections. Through the analysis of the O 1s by XPS [31], it provides a mean to investigate the oxygen-related states in AOS, which can be related to some numerical DOS parameters. Therefore, the bulk and interfacial DOS can be extracted depending on the oxygen ratio during a-IWO deposition. As a result, the proposed physical models and material properties of a-IWO TFT can be validated by TCAD.

First, the DOS of the metal-ions s-band [10,12,23] can be modeled by conduction band tail DOS *g_TA_*(*E*) in Equation (12), and Figure 1b describes the conduction band edge intercept densities *N_TA_* and its decay energy *W_TA_*_._ *N_TA_* is associated with lowering the electron concentration [3]. In this simulation, we assumed *g_TA_*(*E*) was fixed by setting *N_TA_* at 5.0 × 10^19^ cm^−3^·eV^−1^ and *W_TA_* at 0.01 eV [32], because by controlling *N_d_* and *N_C_*, we could observe the change of electron concentration dependent on the oxygen ratio of a-IWO in later sections. On the other hand, the deep DOS of the oxygen *p*-band [23] can be represented by valance band tail DOS *g_TD_*(*E*) in Equation (13) and Figure 1b relating to valence band edge intercept densities *N_TD_* and its decay energy *W_TD_*. Since the transfer I_D_–V_G_ curve was found not affected by deep defect *N_TD_* [32], we assumed *g_TD_*(*E*) was fixed by setting *N_TD_* of 8.0 × 10^20^ cm^−3^·eV^−1^ and *W_TD_* of 0.12 eV for various oxygen ratios of a-IWO [11].
*g_TA_*(*E*) = *N_TA_* exp [(*E* − *E_C_*)/*W_TA_*] (12)
*g_TD_*(*E*) = *N_TD_* exp [(*E_V_* − *E*)/*W_TD_*] (13)

The DOS of oxygen vacancy (V_O_) can be modeled by the Gaussian donor state *g_GD_*(*E*) in Equation (14) and Figure 1b [13], which is dependent on total density *N_GD_*, decay energy *W_GD_*, and peak energy distribution *E_GD_* positioned near *E_C_* by the effect of Madelung potential [3]. Since the amount of V_O_ can be analyzed by XPS for different oxygen ratios of a-IWO, we initially assumed an *N_GD_* of 5.0 × 10^16^ cm^−3^·eV^−1^ with a fixed *W_GD_* of 0.05 eV and an *E_GD_* of 2.95 eV for a 3% oxygen ratio of a-IWO TFT [13]. Then, the I_D_–V_G_ curves were simulated, affected by different *N_GD_* values, as shown in a later section.
*g**_GD_*(*E*) = *N**_GD_* exp {−[(*E* − *E**_GD_*)/*W**_G_**_D_*]^2^} (14)

Other DOS of chemical species relating to hydroxyl (–OH) groups [3], interstitial oxygen (O*_i_*) [33], or metal vacancy [18] in *a*-IGZO can be modeled by Gaussian acceptor state *g_GA_*(*E*) in Equation (15) and Figure 1b, which is dependent on total density *N_GA_*, decay energy *W_GA_*, and peak energy distribution *E_GA_*. To qualitatively analyze the effect of the Gaussian acceptor state in different oxygen ratios of bulk a-IWO, we initially assumed an *N_GA_* of 1.0 × 10^16^ cm^−3^·eV^−1^ with a fixed *W_GA_* of 0.04 eV and *E_GA_* of 0.30 eV [11]. The effect of the *N_GA_* value at the front interface between a-IWO and HfO_2_ on transfer characteristics is discussed as well.
*g**_GA_*(*E*) = *N**_GA_* exp {−[(*E**_GA_* − *E*)/*W**_G_**_A_*]^2^}(15)

The positive gate-bias stress (PGBS) condition causes a positive *V_TH_* shift (∆*V_TH_*), explained by some possible mechanisms such as electron trapping [3] or Joule heating with a hot carrier effect [34], etc. However, in our study, there was no hot carrier effect in the PGBS condition (V_D_ = vs. = 0 V, V_G_ = +2 MV/cm) at room temperature, so we should consider electron trapping after PGBS. Furthermore, it has been reported that a positive *V_TH_* shift (∆*V_TH_*) increases with higher oxygen ratio of AOS TFTs after PGBS [35], which could be ascribed to the greater number of interface traps due to the stronger ion bombardment in the sputtering process [35]. As a result, the geometrical location of interface traps was estimated as the characteristic penetration depth (*d*), which implies charge trapping takes place near the interface between GI and a-IGZO [35]. Moreover, the created oxygen interstitials (O_i_) in the octahedral configuration [18,19] after PGBS, which have been investigated by the first-principles calculation based on density functional theory (DFT). When a large positive V_G_ is applied, the Fermi level (*E_F_*) increases greatly so that the neutral O_i_^0^ states around *E_F_* are occupied by electrons and thus form the negatively charged O_i_^2−^ as chemical reaction formula O_i_^0^ + 2e^−^ ν O_i_^2−^. During PGBS, a stronger defect-lattice electrostatic interaction occurs, and the energy level of a charged state can be lowered below that of the neutral state due to the structural relaxation invoked [29]; such behavior also has been found in amorphous silicon, known as “negative-U” behavior [33,36]. In the oxygen-rich a-IGZO channel, many weakly bonded oxygen can be express as –M_I_⋯O_i_^0^⋯M_II_–, where M_I_ and M_II_ denote metal cations, “–” strong chemical bonds and “⋯” weak chemical bonds [35]. Therefore, it is inferred that weakly bonded oxygen ions are easily ionized under PGBS to form oxygen interstitial (O_i_) defects due to the low defect formation energy *E^f^* [20,36]. The concentration of a native defect in a solid is determined by its formation energy *E^f^* through Equation (16) [20].
*c* = *N_sites_* (−*E^f^*)/*k_B_T*
(16)
where *N_sites_* is the number of sites per unit volume (including different configurations) the defect can be incorporated on, the defect formation energy *E^f^* is dependent on not only the Fermi level *E_F_* but also the chemical potential of the species, *k_B_* is the Boltzmann constant, and *T* is the temperature. Through the DFT calculations, the low defect formation energy of O_i_ in ZnO was found at a high Fermi level (*E_F_*) [14,20]; therefore, one can roughly estimate the possible distribution of O_i_ by observing a high Fermi level (*E_F_*) in AOS TFTs simulation in a later section.

## 4. Results and Discussion

The measured transfer characteristics of a-IWO TFT and the extracted electrical parameters with different oxygen ratios are shown in Figure 2a and Table 1, respectively. When the oxygen ratio increased, the on-state current (*I_ON_*) of the device decreased, the threshold voltage (*V_TH_*) increased, the subthreshold swing (*S.S.*) degraded, and the field-effect mobility (*μ_FE_*) declined correspondingly. The extracted *μ_FE_* dependent on oxygen ratios of a-IWO, in Table 1, were input as the simulation parameters for more accurate material properties. As for the affected transfer characteristics after the PGBS condition (V_D_ = vs. = 0 V, V_G_ = +2 MV/cm) for a certain duration time (2000 s) at room temperature in atmosphere, the stress time dependence of threshold voltage shift (∆*V_TH_*) was extracted, as shown in Figure 2b. Because the transfer characteristics severely degraded with 10% and 13% oxygen ratios after PGBS, the ∆*V_TH_* with only 3% and 7% oxygen ratios are shown; it is noted that the reliability can be improved with low oxygen ratio of a-IWO film.

To verify the mechanisms of positive ∆*V_TH_* with the increasing oxygen ratio, the chemical-bonding states of the a-IWO films were observed using XPS. Because the 4 nm a-IWO active semiconductor was thinner than the 10 nm XPS penetration depth, the prepared sample was 4 nm thick a-IWO film deposited on silicon substrate for analyzing chemical properties of bulk film. In this study, for qualitative numerical analysis, the compositions ratio from XPS O 1s peak could be referenced to deduce the amounts of different density of states (DOS) parameters for AOS simulation. Figure 2c shows the normalized O 1s spectra with increasing oxygen ratios, deconvoluted into three peaks at 529.6, 530.9, and 532.1 eV, respectively. The lowest peak represented the oxygen–metal (O–M) bonding for lattice oxygen (O_L_). The medium peak represented the oxygen vacancy (V_O_), such as the dangling bond or weak bond in oxygen deficient lattice. The highest peak represented the –OH groups, which related to moisture in thin films. When the oxygen ratio increased, the ratios of the area under the spectrum of V_O_ increased from 15.4% to 32.2%, shown in the inset of Figure 2c, where the –OH group also increased from 7.4% to 10.9%.

Apart from XPS O 1s analysis, the stability of a-IWO film can be investigated by XPS W 4f analysis [37]. In the atmosphere, pure tungsten oxide *W*^6+^*O*_3_ is more stable than *W*^4+^*O*_2_, as shown in the following chemical reaction: ${WO}_{3}+H_{2}\to {WO}_{2}+H_{2}O,∆H>0$ [37]. The XPS W 4f spectrum can be deconvoluted into two species (W^6+^ and W^4+^), and therefore the W^6+^ ratio can be regarded as a stability factor. In this study, through XPS W 4f material analysis, the extracted W^6+^ ratios were 83.0, 76.3, 74.9, and 71.0% for 3%, 7%, 10%, and 13% oxygen ratios of a-IWO, respectively. As a result, it could be referred that excess oxygen could produce the unstable *W*^4+^, resulting in an instability, which is consistent with the PGBS results in Figure 2b.

### 4.1. Effect of Dopant Concentration

According to the Poisson’s equation, changing the dopant concentration *N_d_* in device simulation can change the carrier concentration. As a result, the variation of carrier concentration of a-IWO for different oxygen ratios, observed by Hall measurement, can relate to the modulation of dopant concentration *N_d_*. Hence, we simulated how the linear I_D_–V_G_ curves affected by *N_d_* of a-IWO varied from 7.0 × 10^18^ to 7.0 × 10^15^ cm^−3^ in Figure 3a. Although it showed a good fitting on electrical characteristics between measurements and simulations for a 3% oxygen ratio of a-IWO, it is noted that the simulated *V_TH_* shift still could not approach the measurements for 10% and 13% oxygen ratios even with less *N_d_*. For the next analysis of conduction band density of states *N_C_* in a-IWO, we controlled *N_d_* to be 7.0 × 10^18^, 1.0 × 10^18^, 5.0 × 10^15^, and 5.0 × 10^15^ cm^−3^ for 3%, 7%, 10%, and 13% oxygen ratios of a-IWO respectively.

### 4.2. Effect of Conduction Band Density

Although the carrier concentration can be determined by Hall analysis, in device simulation, electron concentration is also affected by conduction band density *N_C_*, according to the related Equations (5), (7), and (9). However, *N_C_* cannot be directly determined by Hall measurement; therefore, *N_C_* values can be numerically deduced for different oxygen ratios of a-IWO. In this section, we simulated how the linear I_D_–V_G_ curves affected by *N_C_* of a-IWO varied from 2.0 × 10^18^ to 2.0 × 10^15^ cm^−3^ in Figure 3b. It was found that the decreased *N_C_* could reduce the simulated S.S. and I_ON_ with a more positive *V_TH_* shift (∆*V_TH_*), and although it was similar to the observed phenomena shown in Figure 2a, nevertheless modulating *N_C_* still could not match with the measurements. In the next section, for analyzing the electronic defect, we controlled *N_C_* to be 2.0 × 10^18^, 1.0 × 10^18^, 3.0 × 10^16^, and 8.0 × 10^15^ cm^−3^ for 3%, 7%, 10%, and 13% oxygen ratios of a-IWO, respectively. In this numerical approach, the physical properties such as band parameters *N_d_*, *N_C_*, or traps, etc., inside a-IWO film can be inferred by fitting with the measured transfer characteristics with different oxygen ratios of a-IWO TFT.

### 4.3. Effect of Gaussian Donor Trap in Bulk a-IWO Film

According to the amount of V_O_ by XPS in Figure 2c, the density of a Gaussian donor trap *N_GD_* can be correlated and assumed to be 5.0 × 10^16^, 5.3 × 10^16^, 5.9 × 10^16^, and 1.1 × 10^17^ cm^−3^·eV^−1^ for 3%, 7%, 10%, and 13% oxygen ratios of a-IWO, respectively. The simulated linear I_D_–V_G_ curves affected by the amount of *N_GD_* are shown in Figure 3c. It was found that more donor traps are ionized as the amount of *p_T_* increased, which was associated with electron generation shown in Equations (6), (8), and Poisson’s equation, so the simulated *V_TH_* changed in a negative shift with increasing *N_GD_* [10]. Hereafter, we elucidated that V_O_ cannot result in a positive *V_TH_* shift (∆*V_TH_*) with increasing oxygen flow during a-IWO deposition. A simple schematic of V_O_ is shown in the inset of Figure 3c. In the next section, for analyzing a Gaussian acceptor trap, we adopted the aforementioned *N_GD_* values for different oxygen ratios of a-IWO to observe the simulated linear I_D_–V_G_ curves affected by the amount of *N_GA_* inside bulk a-IWO.

### 4.4. Effect of Gaussian Acceptor Trap in Bulk a-IWO Film

In the previous XPS O 1s spectra in which the peak intensity at 532.1 eV that could be strictly regarded as the summation from –OH, O_i_, and metal vacancy (V_M_) species, according to the inset table of Figure 2c, its amounts were less than 11% in the total amount of oxygen-related species even with a 13% oxygen ratio of a-IWO. Hence, in this section, we simply denoted it as –OH species by setting the density of the Gaussian acceptor trap *N_GA_* value in bulk a-IWO film, calculated by observing the amount of –OH and V_O_ species at 532.1 and 530.9 eV in Figure 2c, respectively. It was noted that the ratio of V_O_ to –OH was found between 2 and 3. On the other hand, according to the Section 4.3, the density of V_O_ was set between 5.0 × 10^16^ and 1.1 × 10^17^ cm^−3^·eV^−1^, and therefore the density of –OH species could be calculated from 2.5 × 10^16^ to 3.7 × 10^16^ cm^−3^·eV^−1^ with the increasing oxygen ratio. After modulating the range of *N_GA_*, it was found that the simulated I_D_–V_G_ curves were unaffected, as in Figure 3c [11]. In the next section, we adopted the aforementioned bulk *N_GA_* values for different oxygen ratios of a-IWO to observe the simulated linear I_D_–V_G_ curves affected by the density of interfacial Gaussian acceptor trap *N_GA_*. Furthermore, referring to the Poisson’s equation and relations (5), (7), we estimated the acceptor traps not to be ionized as the amount of *n_T_*, which could result from Fermi level position, so the analysis of occupation probability *f*(*E*) of a trap level at energy *E* and Fermi level are presented as well in Section 4.6.

### 4.5. Effect of Gaussian Acceptor Trap at Interface

Because a passivation layer was deposited at the back interface [8], then we assumed that the back interface does not have an interface trap. Furthermore, we assumed the Gaussian acceptor trap to be at the front interface between a-IWO and HfO_2_ as the possible electron recombination, then the simulated linear I_D_–V_G_ curves affected by interface *N_GA_* shown in Figure 3d. To match with the measured I_D_–V_G_ curves, the extracted interface *N_GA_* were 0, 8.0 × 10^12^, 8.0 × 10^13^, and 1.0 × 10^14^ cm^−2^ eV^−1^ for 3%, 7%, 10%, and 13% oxygen ratios of a-IWO, respectively. Consequently, the lower the oxygen ratio of IWO, the lower the interface Gaussian acceptor trap at the front interface was elucidated by simulation, mainly contributing the remarkable experimental electrical characteristics of the a-IWO nanosheet TFT, such as near ideal *S.S.*, small *V_TH_*, and enhanced *I_ON_* [24]. Here the possible species at the front interface could be oxygen interstitial (O_i_) [33], and a simple schematic of O_i_ is shown in the inset of Figure 3d.

### 4.6. Analysis of 1D Fermi Level

To better understand the physics of a-IWO TFT for each oxygen ratio, the analysis of occupation probability *f*(*E*) of either electrons or acceptor traps under equilibrium condition (V_G_ = V_D_ = vs. = 0V) is essential to observe Fermi levels in Figure 4. The different Fermi level positions near front channel of a-IWO TFT can be determined when *f*(*E*) = 0.5 and were dominated mainly by the bulk dopant concentration *N_d_*. Figure 4 indicates the Fermi level position near *E_C_* of only 0.025 eV in a 3% oxygen ratio of a-IWO, representing heavily doped *n*-type material, whereas the observed Fermi level positions were away from *E_C_* for higher oxygen ratios of a-IWO, meaning lightly doped *n*-type material formed consequently. More specifically, the physical mechanisms in a-IWO TFT can be correlated with different DOS from chemical species by analyzing each one-dimensional (1D) energy band diagram, including the Fermi level for different oxygen ratios and bias conditions, and then the 1D and two-dimensional (2D) electron concentration distributions are discussed in later sections.

### 4.7. Analysis of Band Diagram

At equilibrium, the 1D band diagrams from back channel to bottom gate for different oxygen ratios of a-IWO TFT are shown in Figure 5a–d. The 1D electron Fermi level *E_F_*,*_n_* and its corresponding electron concentration distribution inside a-IWO were plotted as well, which can be attributed to the previous effects including bulk dopant concentration *N_d_*, conduction band density *N_C_*, and different chemical DOS distributions. Figure 5a shows the presence of a slight accumulation mode for higher electron concentrations in a 3% oxygen ratio of a-IWO, whereas the depletion modes with flat-bands for higher oxygen ratios of a-IWO depicted in Figure 5b–d, resulted in the reduced electron concentrations. The different interface densities of Gaussian acceptor trap *N_GA_* at the front channel are introduced in Figure 5b–d; it is noted that the *E_F_,_n_* at front interface were below the trap energy *E_GA_*, meaning interface acceptor traps were not ionized as easily as *n_T_* at equilibrium. As a result, at equilibrium, the interface Gaussian acceptor trap was not the main reason for reducing electron concentration inside a-IWO TFT.

Further increasing the gate voltage to the on-state (V_G_ = 7V, V_D_ = 0.1V, vs. = 0V), the focus of 1D band diagrams is on the front channel of a-IWO TFT, depicted in Figure 6a–d for different oxygen ratios. This indicates that the conduction band edges *E_C_* near the front interface were bent by V_G_, and hence the sharp peak electron concentration shifted to the front channel. The various band diagrams represent different degrees of the accumulation mode of a-IWO TFT with different oxygen ratio processes, which were associated with the previous effects including bulk dopant concentration *N_d_*, conduction band density of states *N_C_*, and interface Gaussian acceptor trap density *N_GA_*. The interface Gaussian acceptor trap *g_GA_(E)* at the front channel are introduced in Figure 6b–d; it was found that the *E_F,n_* near the front interface were high and above the interface trap energy *E_GA_*, meaning interface acceptor traps were ionized as *n_T_*, especially in high oxygen ratios of a-IWO, which are associated with electron recombination, with a positive *V_TH_* shift (∆*V_TH_*) observed as a consequence.

### 4.8. Analysis of 2D Distribution of Oxygen Interstitials (O_i_) Formed by Electric Field

In the on-state (V_G_ = 7V, V_D_ = 0.1V, vs. = 0V), large conduction band bending occurs, as observed in Figure 6a–d, resulting in a large electric field (>2 MV/cm) at the front channel, which is like the condition of PGBS. In the Section 4.5, the positive ∆*V_TH_* was analyzed by the interface Gaussian acceptor trap *g_GA_*(*E*), which could be ascribed to created oxygen interstitials (O_i_) either under a large electric field or a strong ion bombardment in sputtering process [35]. Furthermore, through the Equation (16) by DFT, we assumed the formation of O_i_ mainly affected by an increasing large electric field when sweeping V_G_ bias, and consequently we analyzed 2D high *E_F_* distributions to correlate the low formation energy *E^f^*, which means the possible distributions of O_i_ are assessed in Figure 7. The shown contour was confined as band energy difference between quasi-Fermi level (*E_F,n_*) and conduction band edge *E_C_* (−0.1 eV < *E_F,n_* − *E_C_* < 0.1 eV), which indicated that higher Fermi level implies higher concentrations (c) of formed O_i_ at the front channel with deeper penetration depths (*d*) [35] when increasing oxygen ratios during a-IWO sputtering deposition. If we simulated 2D high *E_F_* distributions under PGBS conditions (V_D_ = vs. = 0V, V_G_ = +2 MV/cm), those distributions could be uniform along the lateral (*x*-axis) direction of 2D geometry AOS TFT. In this study, we propose a methodology for monitoring a possibly formed defect as a function of bias condition during numerical analysis of AOS TFTs by knowing the DFT correlation between formation energy *E^f^* of defect and Fermi level (*E_F_*) position.

### 4.9. Analysis of 2D Electron Concentration Distribution

Figure 8a summarizes the four plots of 2D simulated electron distributions inside a-IWO corresponding to oxygen ratios of 3%, 7%, 10%, and 13% in the aforementioned on-state bias condition. Note that the accumulated electrons at the front channel decreased in higher oxygen ratios of a-IWO, which yielded the same results of transfer characteristics in Figure 2a. According to the previous analysis of Fermi level positions and band diagram at the off-state (equilibrium) and the on-state, the positive *V_TH_* shift (∆*V_TH_*) and reduced I_ON_ observed in experimental and simulated transfer characteristics may be associated with the electron trapping at the a-IWO/HfO_2_ interface. However, the procedure of interface electron trapping between the off-state (equilibrium) and the on-state when sweeping V_G_ bias has not been investigated so far, and therefore the amount of trapped electrons at the a-IWO/HfO_2_ interface as a function of V_G_ bias for different oxygen ratios of a-IWO is explored in Section 4.10.

### 4.10. Analysis of Trapped Interface Electron as a Function of V_G_ Bias

Figure 8b shows the changes of the trapped interface electron concentration dependent on V_G_ bias for four different oxygen ratios of a-IWO, which indicates the trapped interface electron concentration increased linearly with increasing V_G_, meaning the traps were filled linearly with electrons when the simulated occupation probability *f*(*E*) linearly approached to the energy position of interface Gaussian acceptor trap energy *E_GA_*. Eventually, the trapped interface electron concentration did not increase any more with increasing V_G_ bias but saturated at a constant value. The linearity range of trapped interface electron concentration increased with increasing oxygen ratio of a-IWO, which could be mainly attributed to the density of the interface Gaussian acceptor trap N_GA._ Furthermore, according to the band diagram at equilibrium in Figure 5, it is noted that the energy difference between *Ec* and *E_F,n_* at the front interface could dominantly determine the amount of the trapped interface electron concentration, which is strongly dependent on V_G_ bias. Therefore, the analysis of band diagrams dependent on operating bias is useful and necessary for observing the impact of oxygen ratio of a-IWO on electrical characteristics.

## 5. Conclusions

The fabricated a-IWO nanosheet TFT with a HfO_2_ high-κ gate insulator (GI) has been demonstrated for its promising electrical characteristics in our previous study. In this study, the more positive *V_TH_* shift and reduced I_ON_ were observed when increasing the oxygen ratio during a-IWO deposition. Through simple materials analysis such as XPS analysis and Hall measurements, clear correlations between different chemical species and the corresponding bulk and interface DOS parameters were systematically deduced by TCAD simulation matching with experimental transfer characteristics, validating the proposed physical models with quantum potential for a-IWO nanosheet TFT. Moreover, the band diagrams, electron concentration, and trapped interface electron concentration as a function of V_G_ were numerically analyzed dependent on different oxygen ratios of a-IWO TFT. The effects of oxygen flow on defects were numerically proved for modulating both bulk dopant concentration *N_d_* and interface density of Gaussian acceptor trap *N_GA_* at the front channel, ascribed to created oxygen interstitials (O_i_) dominating significantly the transfer characteristics of a-IWO TFT. More specifically, we propose a numerical methodology for monitoring the formed O_i_ defect of AOS TFTs as a function of bias condition by knowing the DFT correlation between formation energy *E^f^* of the defect and Fermi level (*E_F_*) position.

## Figures and Tables

**Figure 1 nanomaterials-11-03070-f001:**
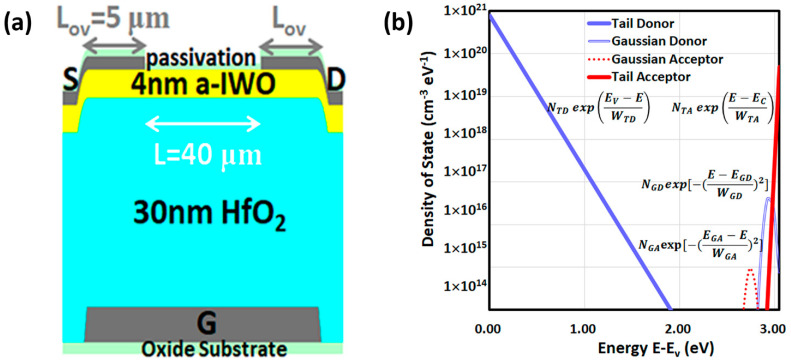
(**a**) The schematic of a BMG a-IWO TFTs. (**b**) The illustrated DOS distribution in a-IGZO.

**Figure 2 nanomaterials-11-03070-f002:**
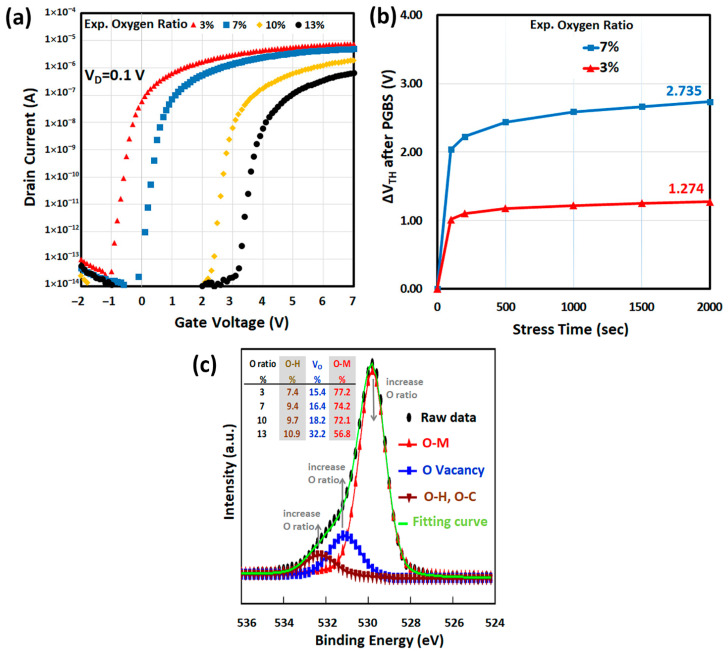
(**a**) The measured transfer characteristics of a-IWO TFT with different oxygen ratios of a-IWO. (**b**) The extracted positive *V_TH_* shift (∆*V_TH_*) dependence of stress time with different oxygen ratios of a-IWO after positive gate-bias stress (PGBS). (**c**) XPS fitting of O 1s with different oxygen ratios of a-IWO.

**Figure 3 nanomaterials-11-03070-f003:**
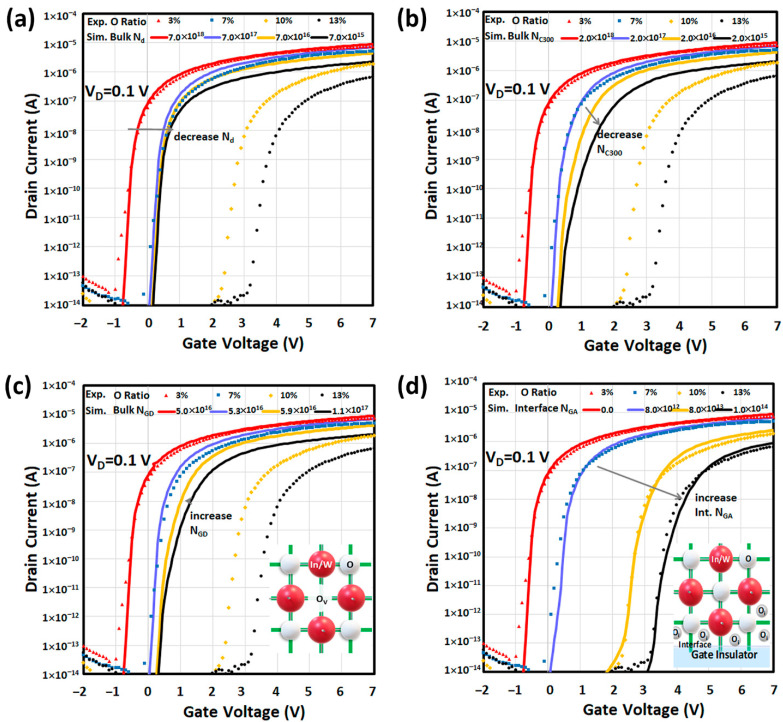
Simulated I_D_–V_G_ curves affected by (**a**) bulk dopant concentration *N_d_*, (**b**) bulk conduction band carrier concentration *N_C_*, (**c**) bulk density of Gaussian donor trap *N_GD_*, and (**d**) front interface density of Gaussian acceptor trap *N_GA_*.

**Figure 4 nanomaterials-11-03070-f004:**
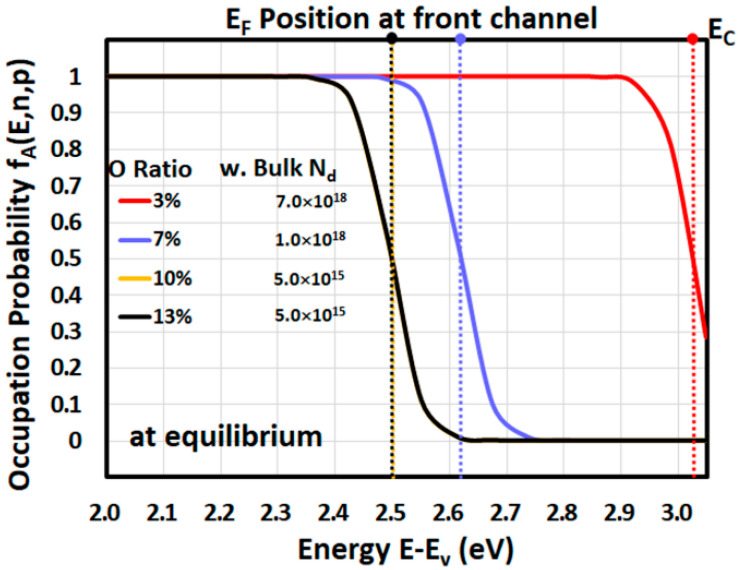
Occupation probability of electrons *f*(*E*) and the Fermi level positions determined when *f*(*E*) = 0.5 at front channel for different oxygen ratios of a-IWO.

**Figure 5 nanomaterials-11-03070-f005:**
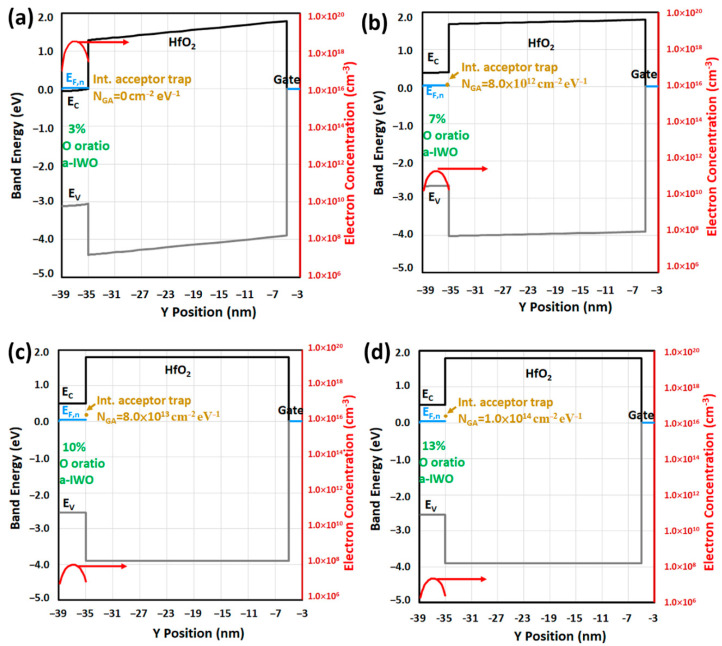
Under equilibrium, 1D band diagrams including electron quasi-Fermi level (*E_F,n_*) and electron concentration for (**a**) 3%, (**b**) 7%, (**c**) 10%, and (**d**) 13% of a-IWO TFT.

**Figure 6 nanomaterials-11-03070-f006:**
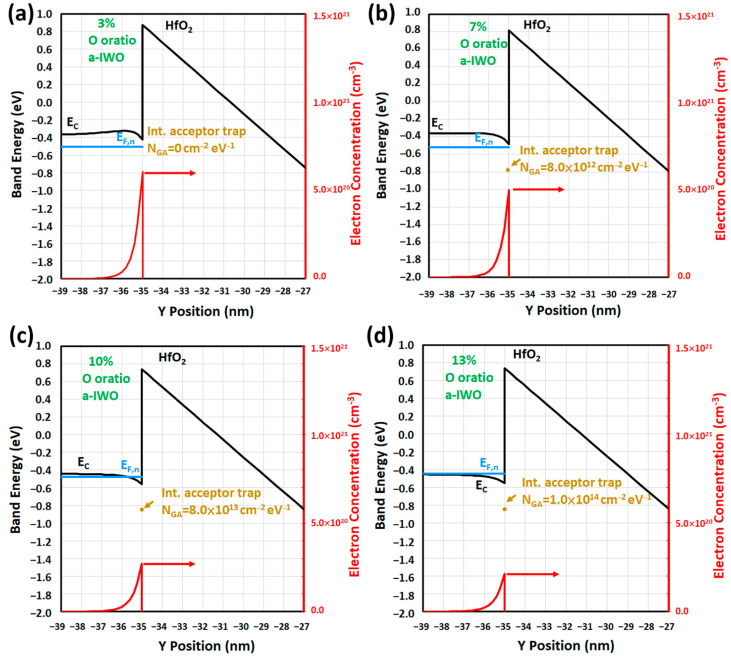
In the on-state (V_G_ = 7V, V_D_ = 0.1V, vs. = 0V), 1D band diagrams including electron quasi-Fermi level (*E_F,n_*) and electron concentration for (**a**) 3%, (**b**) 7%, (**c**) 10%, and (**d**) 13% of a-IWO TFT.

**Figure 7 nanomaterials-11-03070-f007:**
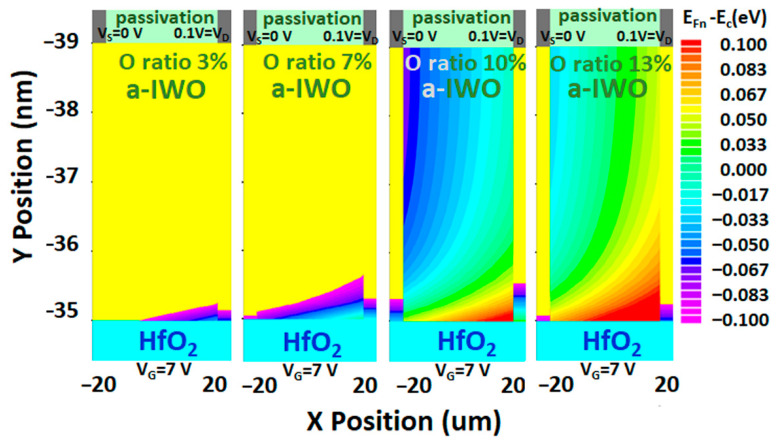
The 2D high Fermi level (*E_F_*) distributions as a function of bias condition for possible distributions of formed oxygen interstitial (O_i_).

**Figure 8 nanomaterials-11-03070-f008:**
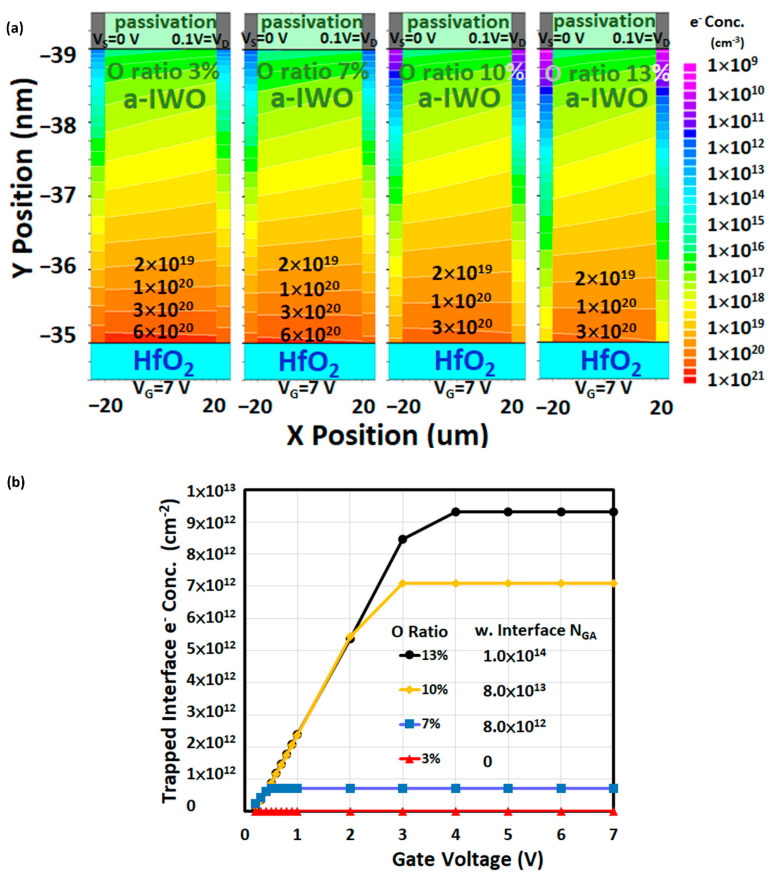
(**a**) The 2D simulated electron distributions inside a-IWO with different oxygen ratios in the on-state condition; (**b**) trapped interface electron concentration dependent on V_G_ bias for four different oxygen ratios of a-IWO TFT.

**Table 1 nanomaterials-11-03070-t001:** Extracted device parameters with different oxygen ratios.

Gas Flow (sccm)		Oxygen Ratio (%)	*V_th_* (V)	*μ_FE_* (cm^2^/Vs)	*S.S.* (mV/dec)	*I_on_*/*I_off_* Ratio
O_2_	Ar					
1	29	3	0.287	24.0	127	1.5 × 10^8^
2	28	7	0.636	19.3	119	9.8 × 10^7^
3	27	10	3.065	12.0	119	3.8 × 10^7^
4	26	13	4.089	5.6	143	1.3 × 10^7^

## Data Availability

The data presented in this study are available on request from the corresponding author. The data are not publicly available due to privacy.

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
