# Peer review of "Numerical Analysis of Oxygen-Related Defects in Amorphous In-W-O Nanosheet Thin-Film Transistor"

_nanomaterials, 2021, doi:10.3390/nano11113070_

Round 1
Reviewer 1 Report
Suggestion Major Revision
Comments
This manuscript describes the numerical simulation of the thin film transistors of metal oxide like as IGZO.
Oxide condition which provides either the positive or negative shift the threshold voltage of the TFTproperty of the IWO.
The sensitive behavior of the TFT device on the oxide condition is well recognized and hard to control thus this report will attract attention for the researcers in this field.
However the referee considers that the follow points should be revised before publication.
1
Over all, although the XPS is a good surface analysis technique, much of the analysis and the conclusion is based on the results of the XPS analysis for the quantative estimation of oxide species. The uthors should state the
1 XPS principle of the recognition of different species from O1s species
2what is the error bars coming from the fitting for deconvolution of the O1s peaks.
2
Line195, the phrases concerning different chemistry depending on the structure is not clear
3
The assignment of OH species to humidity is not rational.
They're intentionally formed as to delete the active site as the forming gas process in MOS device process
4
As the authors stated around line 156, the results of the analysis for
XPS 1s states are used for important parameters. However the description of the=data analysis is far less satisfying.
Why the authors assigned the species to OH related species, for example,has to be explained with much more detail
Author Response
Response to the Reviewer’s Comments
Manuscript: Nanomaterials-1435409
Reviewer #1:
This manuscript describes the numerical simulation of the thin film transistors of metal oxide like as IGZO. Oxide condition which provides either the positive or negative shift the threshold voltage of the TFTproperty of the IWO. The sensitive behavior of the TFT device on the oxide condition is well recognized and hard to control thus this report will attract attention for the researcers in this field. However the referee considers that the follow points should be revised before publication.
- Over all, although the XPS is a good surface analysis technique, much of the analysis and the conclusion is based on the results of the XPS analysis for the quantative estimation of oxide species. The authors should state the
- XPS principle of the recognition of different species from O1s species
Response:
Thanks very much for the referee’s professional advice. The more detailed descriptions about XPS analysis are in Lines 245-265.
“…or qualitative numerical analysis, the compositions ratio from XPS O 1s peak can be referred to deduce the amounts of different density of states (DOS) parameters for AOS simulation.”
- what is the error bars coming from the fitting for deconvolution of the O1s
peaks.
Response:
Thanks very much for the referee’s professional advice. Fig. 2 (b) has been modified clearly to show XPS O 1s spectrum with 3% oxygen ratio, the arrows were to show the variation trends with the increasing oxygen ratio for the three peaks.
- Line195, the phrases concerning different chemistry depending on the structure is not clear
Response:
Thanks very much for the referee’s professional advice. As suggested, the description is modified as following. The more detailed descriptions are in Lines 220-225 as following.
“..a stronger defect-lattice electrostatic interaction occurs, the energy level of a charged state can be lower than that of the neutral state due to structural relaxation invoked, such behavior also has bene found in amorphous silicon.”
- The assignment of OH species to humidity is not rational. They're intentionally formed as to delete the active site as the forming gas process in MOS device process
- As the authors stated around line 156, the results of the analysis for XPS 1s states are used for important parameters. However the description of the=data analysis is far less satisfying. Why the authors assigned the species to OH related species, for example, has to be explained with much more detail
Response:
Thanks very much for the referee’s professional advice. To answer above question 3 and 4, the description is in section 4.3 as following.
“Strictly speaking, the peak intensity at 532.1 eV in XPS O 1s spectrum that could be regarded as the summation from -OH, Oi and metal vacancy (VM) species. However, its amounts were less than 11% from the inset table of Figure 2 (b), in this section we simply denoted it as -OH species by setting the density of Gaussian acceptor trap NGA in bulk a-IWO film calculated by observing the amount of -OH and VO species at 532.1 and 530.9 eV in Fig. 2 (b), respectively. it’s noted that the ratio of VO to -OH was be found between 2 and 3. On the other hand, according to the section 4.2, the density of VO was set between 5.0´1016 and 1.1´1017 cm-3·eV-1, and therefore the density of -OH species can be calculated from 2.5´1016 to 3.7´1016 cm-3·eV-1 with the increasing oxygen ratio. And then modulating the range of NGA, it’s found the simulated ID–VG curves were unaffected alike in Figure 3 (c) [11].
Reviewer 2 Report
1. Please provide more background in the introduction on the research challenges associated with the particular materials and methods of this work.
2. Figure 2 and 3 show the "normalized current". Please explain what this means. Typically, normalized current in a transistor means the current density through the gate, i.e. Id divided by the gate width. However these figures show current values, not densities, judging by the unit.
3. It would be good to show IdVd characteristics of the transistors.
4. Have you considered the impact of deeper defect states, so called border traps?
Author Response
Response to the Reviewer’s Comments
Manuscript: Nanomaterials-1435409
Reviewer #2:
- Please provide more background in the introduction on the research challenges associated with the particular materials and methods of this work.
Response:
Thanks very much for the referee’s professional advice. As suggested, the descriptions are added in Lines 60 -62 and Lines 74-77as following.
“…for developing and exploring vast AOS materials, theoretical DFT calculations would be time-exausting, utilizing TCAD numerical analysis together with simple material analysis could an alternative and prompt solution to AOS device.”
“…we investigate experimentally the electrical properties of nanosheet (NS) junctionless a-IWO TFT dependent on different oxygen flows during a-IWO deposition, then deduce numerically the correlation among the chemical species, materials properties, DOS and band diagrams by TCAD [25].”
- Figure 2 and 3 show the "normalized current". Please explain what this means. Typically, normalized current in a transistor means the current density through the gate, i.e. Id divided by the gate width. However these figures show current values, not densities, judging by the unit.
Response:
Thanks very much for the referee’s kind reminder. The Y-axis should be Drain current in Ampere unit. Figure 2 and 3 have been modified in the manuscript.
- It would be good to show IdVd characteristics of the transistors.
Response:
Thanks very much for the referee’s professional advice. Below is IdVd cures dependent on different oxygen ratio, it’s noted that an increased source/drain (S/D) contact resistance (RC) occurred in excess oxygen process. In our previous study [24], an excellent low RC was achieved with thinner gate insulator (GI) and low oxygen ratio condition in a-IWO film. Therefore, the IdVd curves are not to be shown in this paper.
- Have you considered the impact of deeper defect states, so called border traps?
Response:
Thanks very much for the referee’s professional advice. Yes, the descriptions are in Lines 193-197 as following.
“…the deep DOS of oxygen p-band [23] can be represented by valance band tail DOS gTD(E) in equation (13) and Figure 1 (b) relating to valence band edge intercept densities NTD and its decay energy WTD. Since the transfer ID–VG curve was found not affected by deep defect NTD [32], so we assumed gTD(E) was fixed by setting NTD of 8.0´1020 cm-3·eV-1 and WTD of 0.12 eV for various oxygen ratios of a-IWO [33]”

Reviewer 3 Report
In the present manuscript, Fan and co-workers have demonstrated the influence of the nature, position and type of the defects in an amorphous indium tungsten oxide-based TFT fabricated via sputter deposition. The authors focus on the influence of oxygen on the resultant defects generated within the semiconductor. Increased acceptor-like defects at the semiconductor/dielectric interface were concluded as the main cause of the reduced On current, a positive shift in the threshold voltage and reduced sub-threshold swing. The TCAD simulations successfully corroborate with the findings that reduced oxygen minimizes interfacial defects, thereby, delivering the best TFT performance. Additional, 2D distribution of oxygen interstitials and general electron concentration have also been discussed in detail. The publication is certainly of good value to the readership of Nanomaterials, specifically to the oxide TFT fraternity. However, prior to publication, the authors should address the following points
Specific Revision:
- The authors assume that the DOS of a-IWO and a-IGZO are the same. Although the simulations fit the experimental observations, the authors should clearly mention or provide a reasonable explanation for this assumption as the authors themselves state the bond dissociation energy for W is rather different than that of Ga and Zn in a typical IGZO TFT (also different material compositions)
For e.g. based on observed carrier concentration, bandgap of the material etc.
- The authors should include a neat schematic showing the different nature-position of defects investigated/simulated for measured TFTs. Specifically relating to Fig.3. This would boost the understanding of the general readership.
- Have the authors measured the XPS depth spectra (or Grazing Angle XPS for rather thin films) to observe a difference in the nature of the vacancy defects at the interface and in the bulk for the different samples. This would also support their claim of interfacial defects, rather than bulk is largely responsible for the TFT performance, since the TFT performance is heavily correlated with the interfacial defects.
For e.g.
https://iopscience.iop.org/article/10.1149/MA2018-02/36/1206/meta
https://pubs.rsc.org/en/content/articlehtml/2017/tc/c7tc03724d
The authors should also include more literature discussion on interfacial and bulk defects for IGZO or other oxide TFT based systems.
- Since the authors include discussions concerning the change of electron concentration trapped at the interface upon gate bias, they should certainly include positive bias gate stress (PBGS) measurements wherein significant shift in the higher O2 This will further validate the findings from the simulation and confirm that the acceptor interfacial traps generated for high O2 concentration TFT are indeed responsible for the positive Vth and reduced On current.
https://ieeexplore.ieee.org/document/5657252
General revisions:
- Please specify the passivation layer used, since the damage to the front end can vary drastically based on whether it is organic or inorganic passivation and the type of deposition used.
- The Yellow coloured font is not clearly visible in Fig. 5 and 6. Please change to high contrast colour.
- The labelling of the S/D and passivation should be consistent and systematic in Fig.7.
- In Fig.8a, it would be helpful to flip the direction of the colour indicator of the e- conc. scale bar for the easier association.
Author Response
Response to the Reviewer’s Comments
Manuscript: Nanomaterials-1435409
Reviewer #3:
In the present manuscript, Fan and co-workers have demonstrated the influence of the nature, position and type of the defects in an amorphous indium tungsten oxide-based TFT fabricated via sputter deposition. The authors focus on the influence of oxygen on the resultant defects generated within the semiconductor. Increased acceptor-like defects at the semiconductor/dielectric interface were concluded as the main cause of the reduced On current, a positive shift in the threshold voltage and reduced sub-threshold swing. The TCAD simulation ssuccessfully corroborate with the findings that reduced oxygen minimizes interfacial defects, thereby, delivering the best TFT performance. Additional, 2D distribution of oxygen interstitials and general electron concentration have also been discussed in detail. The publication is certainly of good value to the readership of Nanomaterials, specifically to the oxide TFT fraternity. However, prior to publication, the authors should address the following points
Specific Revision:
- The authors assume that the DOS of a-IWO and a-IGZO are the same. Although the simulations fit the experimental observations, the authors should clearly mention or provide are reasonable explanation for this assumption as the authors themselves state the bond dissociation energy for W is rather different than that of Ga and Zn in a typical IGZO TFT (also different material compositions)
For e.g. based on observed carrier concentration, bandgap of the material etc.
Response:
Thanks very much for the referee’s professional advice. The description is modified to be in Line 169 as following.
Replacing Ga and/or Zn in InO-based semiconductors, such as InTiO, InWO etc, that is an alternative way to suppress VO for reducing the instability [4-8], which still maintains the electronic configurations of AOS.
- The authors should include a neat schematic showing the different nature-position of defects investigated/simulated for measured TFTs. Specifically relating to Fig.3. This would boost the understanding of the general readership.
Response:
Thanks very much for the referee’s professional advice. Two neat schematic for Vo and interface Oi were added in Figure 3 (c) and (d).
- Have the authors measured the XPS depth spectra (or Grazing Angle XPS for rather thin films) to observe a difference in the nature of the vacancy defects at the interface and in the bulk for the different samples. This would also support their claim of interfacial defects, rather than bulk is largely responsible for the TFT performance, since the TFT performance is heavily correlated with the interfacial defects.
For e.g.
https://iopscience.iop.org/article/10.1149/MA2018-02/36/1206/meta
https://pubs.rsc.org/en/content/articlehtml/2017/tc/c7tc03724d
The authors should also include more literature discussion on interfacial and bulk defects for IGZO or other oxide TFT based systems.
Response:
Thanks very much for the referee’s professional advice. No, the analysis of XPS depth spectra (or Grazing Angle XPS) hasn’t been applied for ultra-thin a-IWO film. We’ll do an interfacial material analysis to strongly support our numerical methodology in next study.
- Since the authors include discussions concerning the change of electron concentration trapped at the interface upon gate bias, they should certainly include positive bias gate stress (PBGS) measurements wherein significant shift in the higher O. This will further validate the findings from the simulation and confirm that the acceptor interfacial traps generated for high O concentration TFT are indeed responsible for the positive Vth and reduced On current.
https://ieeexplore.ieee.org/document/5657252
Response:
Thanks very much for the referee’s professional advice. The more significant positive Vth shift observed with higher oxygen ratio of a-IWO after PBGS, which could be explained by the acceptor interfacial traps for next publication.
Besides, in this study, the added descriptions in Lines 263-270 to show the stability results.
“…XPS W 4f spectrum can be deconvoluted into two species (W6+ and W4+) and therefore W6+ ratio can be regarded as a stability parameter. In this study, through XPS W 4f material analysis, the extracted W6+ ratio were 83.0, 76.3, 74.9 and 71.0 % for 3%, 7%, 10% and 13% oxygen ratios of a-IWO, respectively. As a result, it could be deduced that excess oxygen could produce the unstable W4+ which would result in an instability, that will be numerically analyzed by TCAD in later section.”
General revisions:
- Please specify the passivation layer used, since the damage to the front end can vary drastically based on whether it is organic or inorganic passivation and the type of deposition used.
Response:
Thanks very much for the referee’s professional advice. As suggested, the descriptions are add in Lines 94-96 as following.
“an organic passivation layer was deposited by spin coating at back channel of TFTs and annealed at 150℃ for 30 minutes to ensure a high reliability [4,7]”
Besides, different passivation layers including organic or inorganic materials have been investigated, it’s found that an organic material deposited by spin coating, which has a better passivation effect. The detailed study is prepared for next publication recently.
- The Yellow coloured font is not clearly visible in Fig. 5 and 6. Please change to high contrast colour.
Response:
Thanks very much for the referee’s professional advice. Fig. 5 and 6 have been modified with gold coloured font instead of yellow one.
- The labelling of the S/D and passivation should be consistent and systematic in Fig.7.
Response:
Thanks very much for the referee’s professional advice. Fig. 7 has been modified in the manuscript.
- In Fig.8a, it would be helpful to flip the direction of the colour indicator of the e- conc. scale bar for the easier association.
Response:
Thanks very much for the referee’s professional advice. Fig. 8 a has been modified in the manuscript.

Round 2
Reviewer 1 Report
The authors revised the main text according to the referee's concerns. The contents are much improved and the referee considers it is adequate to publish this article as it is.
Author Response
The authors revised the main text according to the referee's concerns. The contents are much improved and the referee considers it is adequate to publish this article as it is.
Response:
Thanks very much for the referee’s professional advice. As suggested, the modified manuscript is uploaded for reference.

Reviewer 3 Report
Based on the general revision comments:
- Please specify the passivation layer used, since the damage to the front end can
vary drastically based on whether it is organic or inorganic passivation and the type of deposition used.
Response:
Thanks very much for the referee’s professional advice. As suggested, the descriptions are add in Lines 94-96 as following.
“an organic passivation layer was deposited by spin coating at back channel of TFTs and annealed at 150℃ for 30 minutes to ensure a high reliability [4,7]”
The authors have cited two references, one by themselves, which has a SiO2 passivation layer and the other focuses on a bilayer oxide system, which has no passivation layer and contains an all-inorganic layer system. Both of the mentioned citations do not mention any organic passivation layer, which makes their methods incomplete to provide the fabrication steps used in their study Please clearly mention the organic compound, which was spin-coated. (For e.g. PMMA, PVA etc. ) with relevant citations or avoid unnecessary citations completely.
Based on the specific revision comments:
- Since the authors include discussions concerning the change of electron concentration trapped at the interface upon gate bias, they should certainly include positive bias gate stress (PBGS) measurements wherein significant shift in the higher O. This will further validate the findings from the simulation and confirm that the acceptor interfacial traps generated for high O concentration TFT are indeed responsible for the positive Vth and reduced On current.
https://ieeexplore.ieee.org/document/5657252
Response:
Thanks very much for the referee’s professional advice. The more significant positive Vth shift observed with higher oxygen ratio of a-IWO after PBGS, which could be explained by the acceptor interfacial traps for next publication.
Besides, in this study, the added descriptions in Lines 263-270 to show the stability results.
“…XPS W 4f spectrum can be deconvoluted into two species (W6+ and W4+) and therefore W6+ ratio can be regarded as a stability parameter. In this study, through XPS W 4f material analysis, the extracted W6+ ratio were 83.0, 76.3, 74.9 and 71.0 % for 3%, 7%, 10% and 13% oxygen ratios of a-IWO, respectively. As a result, it could be deduced that excess oxygen could produce the unstable W4+ which would result in an instability, that will be numerically analyzed by TCAD in later section.”
The PBGS measurements have not been performed as requested. Rather the authors have provided an explanation based on the cationic states of the tungsten species with the a-IWO film. However, multiple phenomena occur at the interface leading to the Vth shift apart from acceptor interfacial traps at a TFT device level, such as source-drain joule heating among other effects. (Refer summary of 10.2 of the Ref. 3 cited in this present version of the manuscript: https://doi.org/10.1002/pssa.201800372)HEnce, this should be clearly tested, since the authors already have fabricated the devices with the available Semiconductor parameter analyzer. Else, the authors should eliminate the discussion based on the bias stability of devices and include it in their future publication with verification to their claim.
- Have the authors measured the XPS depth spectra (or Grazing Angle XPS for rather thin films) to observe a difference in the nature of the vacancy defects at the interface and in the bulk for the different samples. This would also support their claim of interfacial defects, rather than bulk is largely responsible for the TFT performance, since the TFT performance is heavily correlated with the interfacial defects.
For e.g.
https://iopscience.iop.org/article/10.1149/MA2018-02/36/1206/meta
https://pubs.rsc.org/en/content/articlehtml/2017/tc/c7tc03724d
The authors should also include more literature discussion on interfacial and bulk defects for IGZO or other oxide TFT based systems.
Response:
Thanks very much for the referee’s professional advice. No, the analysis of XPS depth spectra (or Grazing Angle XPS) hasn’t been applied for ultra-thin a-IWO film. We’ll do an interfacial material analysis to strongly support our numerical methodology in next study.
XPS penetration depth is about 10nm, which is more than double the thickness of the 4nm IWO layer. This implies that the analyzed spectra also contain significant contributions from the underlying HfO2, which does not have an atomic–level smoothness and in turn, will also vary in its own O1s contributions. The author's previous publications contained a 30nm a-IWO which correctly ensures a reliable XPS analysis. (Ref. 4) The authors need to justify that the O1s spectra are purely comprised of contributions from the 4nm a-IWO, which can be easily cross-verified in the survey spectra of the measurements (where contributions from Hf will be visible) which was also the reason for specific XPS analysis. If the authors can provide a clear, well-grounded scientific explanation for avoiding it, the authors should include possible ambiguity of the further research as they suggested and centre some short discussion around XPS analysis based on XPS investigations of ultra-thin layers based on literature and citations provided and accordingly mention their claims are open to further research.
Author Response
Based on the general revision comments:
1. The authors have cited two references, one by themselves, which has a SiO2 passivation layer and the other focuses on a bilayer oxide system, which has no passivation layer and contains an all-inorganic layer system. Both of the mentioned citations do not mention any organic passivation layer, which makes their methods incomplete to provide the fabrication steps used in their study Please clearly mention the organic compound, which was spin-coated. (For e.g. PMMA, PVA etc.) with relevant citations or avoid unnecessary citations completely.
Based on the specific revision comments:
Response:
Thanks very much for the referee’s professional advice. The descriptions were modified in Line 93-95 as following.
“Finally, an organic epoxy-based negative photoresist organic material, SU-8, was deposited by spin coating at back channel of TFTs as a passivation layer and then annealed at 150 ˚C for 30 minutes to ensure a high reliability [8].”
Based on the specific revision comments:
4. The PBGS measurements have not been performed as requested. Rather the authors have provided an explanation based on the cationic states of the tungsten species with the a-IWO film. However, multiple phenomena occur at the interface leading to the Vth shift apart from acceptor interfacial traps at a TFT device level, such as source-drain joule heating among other effects. (Refer summary of 10.2 of the Ref. 3 cited in this present version of the manuscript:https://doi.org/10.1002/pssa.201800372) HEnce, this should be clearly tested, since the authors already have fabricated the devices with the available Semiconductor parameter analyzer. Else, the authors should eliminate the discussion based on the bias stability of devices and include it in their future publication with verification to their claim.
Response:
Thanks very much for the referee’s professional advice. The measured PBGS results are added in Line 251-257 as following.
“As for the affected transfer characteristics after PGBS condition (VD=VS= 0V, VG= +2 MV/cm) for a certain duration time (2000 sec) at room temperature in atmosphere, the stress time dependence of threshold voltage shift (∆VTH) was extracted as shown in Figure 2 (b). Because the transfer characteristics severely degraded with 10 % and 13% oxygen ratios after PGBS, the ∆VTH with only 3% and 7% oxygen ratios were shown, it’s noted that the reliability can be improved with low oxygen ratio of a-IWO film”
The descriptions of possible PGBS mechanisms are added in Line 210-216 as following.
“The positive gate-bias stress (PGBS) condition causes positive VTH shift (∆VTH) explained by some possible mechanisms, such as electron trapping [3] or Joule heating with hot carrier effect [34] etc. However, in our study, there would be no hot carrier effect in PGBS condition (VD=VS= 0V, VG= +2 MV/cm) at room temperature, so we should consider electron trapping after PGBS. Besides, it has been reported that positive VTH shift (∆VTH) increases with higher oxygen ratio of AOS TFTs after PGBS [35], which could be ascribed to the more interface traps due to the stronger ion bombardment in sputtering process [36].”
3. XPS penetration depth is about 10nm, which is more than double the thickness of the 4nm IWO layer. This implies that the analyzed spectra also contain significant contributions from the under lying HfO2, which does not have an atomic–level smoothness and in turn, will also vary in its own O1s contributions. The author's previous publications contained a 30nm a-IWO which correctly ensures a reliable XPS analysis. (Ref. 4) The authors need to justify that the O1s spectra are purely comprised of contributions from the 4nm a-IWO, which can be easily cross-verified in the survey spectra of the measurements (where contributions from Hf will be visible) which was also the reason for specific XPS analysis. If the authors can provide a clear, well-grounded scientific explanation for avoiding it, the authors should include possible ambiguity of the further research as they suggested and centre some short discussion around XPS analysis based on XPS investigations of ultra-thin layers based on literature and citations provided and accordingly mention their claims are open to further research.
Response:
Thanks very much for the referee’s professional advice. The detailed XPS descriptions are modified in Line 258-262 as following.
“To verify the mechanisms of positive ∆VTH with the increasing oxygen ratio, the chemical-bonding states of the a-IWO films were observed using XPS. Because 4 nm thick a-IWO active semiconductor was thinner than 10 nm of XPS penetration depth, the prepared sample was 4 nm a-IWO film deposited on silicon substrate for analyzing chemical properties of bulk film.”
